# Experimental Strain Measurement Approach Using Fiber Bragg Grating Sensors for Monitoring of Railway Switches and Crossings

**DOI:** 10.3390/s21113639

**Published:** 2021-05-24

**Authors:** Abdelfateh Kerrouche, Taoufik Najeh, Pablo Jaen-Sola

**Affiliations:** 1School of Engineering and the Built Environment, Edinburgh Napier University, 10 Colinton Road, Edinburgh EH10 5DT, UK; p.sola@napier.ac.uk; 2Department of Civil, Environmental and Natural Resources Engineering, Luleå University of Technology, SE-971 87 Luleå, Sweden; taoufik.najeh@ltu.se

**Keywords:** switches and crossings, S&C, wear effects, fiber Bragg grating, FBG, sensors, strain measurements

## Abstract

Railway infrastructure plays a major role in providing the most cost-effective way to transport freight and passengers. The increase in train speed, traffic growth, heavier axles, and harsh environments make railway assets susceptible to degradation and failure. Railway switches and crossings (S&C) are a key element in any railway network, providing flexible traffic for trains to switch between tracks (through or turnout direction). S&C systems have complex structures, with many components, such as crossing parts, frogs, switchblades, and point machines. Many technologies (e.g., electrical, mechanical, and electronic devices) are used to operate and control S&C. These S&C systems are subject to failures and malfunctions that can cause delays, traffic disruptions, and even deadly accidents. Suitable field-based monitoring techniques to deal with fault detection in railway S&C systems are sought after. Wear is the major cause of S&C system failures. A novel measuring method to monitor excessive wear on the frog, as part of S&C, based on fiber Bragg grating (FBG) optical fiber sensors, is discussed in this paper. The developed solution is based on FBG sensors measuring the strain profile of the frog of S&C to determine wear size. A numerical model of a 3D prototype was developed through the finite element method, to define loading testing conditions, as well as for comparison with experimental tests. The sensors were examined under periodic and controlled loading tests. Results of this pilot study, based on simulation and laboratory tests, have shown a correlation for the static load. It was shown that the results of the experimental and the numerical studies were in good agreement.

## 1. Introduction

### 1.1. Railway Switches and Crossings Monitoring

There are, primarily, two main approaches in the railway condition monitoring field [1,2,3]: direct and indirect. Direct methods are based on digital photography installed on board, or the wayside, and target predefined locations of the railway. Indirect approaches evaluate degradation using a data acquisition system to collect different types of signals produced by the wheel–rail interaction. These signals can be vibration signals, pressure, normal force, and speed. The collected signals will feed signal processing tools to extract any feature that reflects railway health conditions. Switches and crossings (S&C) are significant components of the railway infrastructure due to their important contributions to traffic delay and maintenance costs. S&C are mechanical installation systems used in complex railway networks to guide trains from one track to another and to support vehicles safely when passing to the turnout direction or through the direction. S&C are critical systems in the railway network, and are required to be reliable in order to prevent delays or fatal accidents [4]. Maintaining S&C systems for preventive maintenance, to allow higher flexibility across the entire rail network, is very expensive. In Sweden, there are about 14,000 S&C systems, and the average lifespan of an S&C system is less than 3 years [5]. This can lead to high demand in delivering new approaches, as well as prognostic health management (PHM) tools to predict changes in the behaviors of the structures of S&C systems at early stages of degradation. In the UK, in 2008, it was estimated that S&C systems cost the rail infrastructure owners and operators over GBP 220 million in maintenance [6].

A common S&C system consists of several parts, such as the mechanical switch layout and the electric/electronic control systems. S&C systems have numerous variations in their complex designs, due to variables such as diversion routes and switch lengths. The crossing part (more precisely, the frog section) is a crucial section that is required to guide a train when it is passing from the middle part to the turnout or through the direction of the S&C (Figure 1).

The common failure modes in S&C systems are due to irregularities and discontinuities of the railway paths, which interrupt the wheel–rail contact and cause disruption of the contact trajectories. Figure 2a shows a picture of the frog of an S&C system with the wear effects. As can be seen, most of the wear is on the left side or the right side of the frog. This is quite reasonable because of the wheel’s flange and the frog interaction. This interaction area is ruled by the dimension of the flange (largest diameter of the wheel) and the high of the nose. In a real case scenario, the wear is mainly distributed at the top side of the frog, as shown in Figure 2b. 

Failure of the frog point, due to wear caused by friction and high impact, is one of the causes of full S&C system failure [7,8]. These parts often fail due to wear as they are exposed to harsh environments and contact conditions between the wheel and rail. A review of typical rail damage due to wear was conducted by Bayer et al. [9]. The wear observed by Bayer can be divided into different categories, abrasion (causing high friction levels between the surfaces), impact (causing plastic deformation, and cracks), and delamination (causing material removal from the surface). The side abrasion wear in the switch panels has been the subject of research studies, to understand the loading and degradation of S&C [10]. 

There is limited knowledge in regards to wear during operation, and its impact on the life cycle of in-service S&C. This is due to variations in the contact dynamic forces at different contact locations between the wheel and the rail. However, recent developments in sensor technology and Internet of Things (IoT) devices provide new opportunities for railway owners to reduce visual inspection and maintenance costs by using in-field monitoring systems. Moussa Hamadache et al. have discussed, in a research overview, the sensors used for railway system monitoring with comprehensive state-of-the-art fault detection and diagnosis techniques for railway S&C systems [11]. Among other sensors, strain gauges were used to understand the material reaction and the degradation of S&C due to in-service loadings. 

Strain gauges on crossings were installed in 1978 by Boutle et al. comparing bending strains of two crossings. The results showed that the two designs generated very similar values. However, the strain reduced at higher speeds, but similar strains were measured with a heavier locomotive [12,13]. Uwe Oßberger et al. showed a fixed point crossing equipped with an experimental tool steel crossing nose with strain gauges. Signals from these sensors were recorded in regular intervals to establish a database to develop signal-based condition monitoring [14]. Strain gauges were attached to support brackets and to three positions of the check rail to study the effect of train velocity [15]. 

In recent years, the application of optical fiber sensors (such as Mach–Zehnder, Michelson, or Fabry–Perot) on railway health monitoring was investigated [16]. The Mach–Zehnder sensor measures the variation of the phase shift between two collimated beams derived by splitting light from a single light source [17,18]. The Michelson interferometer device measures the optical path difference between two splitting laser beams [19,20]. The Fabry–Perot sensor consists of a single-mode optical fiber comprising two mirror surfaces. The device measures the frequency shift introduced by a change in the optical path length between these two mirrors [21,22]. These sensors are widely used for many applications, such as structural health monitoring, and showed distinctive advantages (e.g., small size, lightweight, and embedding capability) [23,24]. In this study, a measuring method using fiber Bragg grating (FBG) optical fiber sensors for the bi-directional strain method is presented. This method will allow future development of a rail strain monitoring device for long-term and multi-point measurements, providing relevant in-field experimental tools for S&C automatic condition monitoring systems.

### 1.2. FBG Sensors Solution

Railway infrastructure monitoring can provide valuable information about displacement and track geometry, allowing better understanding of the performance and the condition of the track. Many technologies for monitoring railway tracksides were applied and proved to provide accurate measurements with different types of sensors and techniques, such as acoustic [25], electromagnetic acoustic transducer [26], or image processing [27]. However, most of these sensors are (likely) to be expensive, and their deployment in the field is, relatively, very limited. Therefore, a measurement technique that can continuously monitor railway S&C in real-time and send out alert signals if they are likely to cause damage to the railway structure, is in high demand. The work presented herewith aims to tackle this issue, utilizing an FBG optical sensor-based design that would monitor the early stage of degradation, which can be detected by measuring strain continuously of the frog. An FBG sensor utilizes glass/plastic fiber carrying light, and possesses several advantages, such as easy handling, integration, simplicity, and its small size. Furthermore, its linear capability, high resolution, accuracy, and wavelength multiplexing capability, makes an FBG sensor the ideal candidate for this type of measurement.

The objective of the work presented in this paper is to explore the potential of FBG sensors to monitor the wear on the railway S&C system. This solution could be employed in the field as an early warning device to prevent complete failure. The works by Lee et al. [28] present a system using FBG sensors to count axles and, thus, derailment detection, identification of different train types, and speed detection. Although the work presented here is, to some extent, similar to that by Lee et al. [28], it differs in terms of focus on the S&C parts of the railway infrastructure, and the orientation of the sensors, as well as the testing techniques. Filograno et al. [29] installed many FBG sensors in the rails, in a straight section of a high-speed line to detect defective wheels. The experimental set-up identified few defects on wheels (confirmed by visual inspections). Roveri et al. [30] carried out experimental tests by installing an FBG sensor array system for a six-month period. Fifty FBG sensors were used for health monitoring by measuring strain applied on both wheels and rails. This technique generated a large amount of data, and the algorithms helped to estimate the wear on the rails and wheels. However, the work presented in this paper is innovative and it is based on laboratory-scale experiments to emphasis the use of FBG sensors for monitoring the wear on railway S&C. This work shows a feasibility study using FBG sensors on a metallic crossing to determine the wear effects and the suitability of these sensors for such applications. 

The FBG sensors used in this paper were fabricated in-house at City University London using the phase mask technique. The 248 nm KrF excimer laser, with pulse energy of 12 mJ and a pulse frequency of 200 Hz, was used for UV radiation. Then, the sensors were placed at 185 °C for 3 h to allow temperature stability during the tests. The wavelengths of the phase masks were used within the detection range of the Micron Optics SM125 interrogator system. The central wavelengths of the three FBG sensors used in these experimental tests were as follows: 1548, 1532, and 1542 nm. The reflectivity with the used set-up was about 99%. The experimental tests presented in this paper were carried out in a laboratory environment, at room temperature of 22 °C.

## 2. Methods and Measurement

### 2.1. Simulation Study

The finite element approach is a well-known numerical method used for many engineering applications. Over the last decades, it was used to study physical behaviors, such as strain measurements, stress analysis, and deformation of mechanical structures of many types of complex problems. Briscoe and Chateauminois [31] studied strains developed in a metal-polymer contact under different loading arrangements. They determined friction coefficient under torsion and sliding motion. Simandjuntak et al. [32] studied fatigue crack closure of a corner crack. Four strain gauges were used around the crack tips to determine the opening stress levels and compliance curves. Kanehara and Fujioka [33] developed a novel approach to measure wheel load and lateral force by improving a conventional method. Seven strain gauges were used to detect compressive strain generated by wheel load, and four strain gauges were placed on the disk to measure surface strain introduced by disk bend.

In this work, a 3D meshed model, as shown in Figure 3, was designed to simulate the frog part of the S&C for strain measurement analysis to efficiently and accurately predict its behavior under various inputs, and to optimize its performance and geometry to meet certain design and performance requirements. A detailed finite element model of the fixed frog was generated using the structural finite element simulation add-in package of SolidWorks, which is a powerful piece of software used to design and build mechatronics systems. A non-linear static study replicating the same features of the experimental test carried out in the lab was developed for validation. In this study, it was considered that the part was made of a typical structural stainless steel, such as AISI 304, with Young’s modulus of 190 GPa, a Poisson’s ratio of 0.29, and a density of 8000 kg/m^3^. A high-quality tetrahedral mesh with an average element size of 5.45 mm, composed of 11,103 nodes and 7116 elements obtained from the pertinent mesh independence study was employed in the simulation study. The part shown in Figure 3a was subject to a nominal force of 5 kN that was applied in 2 steps over 75 s at room temperature of 22 °C as is displayed in Figure 3b replicating the second linear loading test studied in the lab, which corresponds to the worst case scenario for validation. In this simulation study, these details were selected to have a comparison point with the experimental tests. In the laboratory, the maximum load was set to a maximum of 5 kN using the Instron 3367 30 kN Universal Strength Tensile Testing loading machine for safety, to keep the clamps used to fix the metallic part in a stable position. The time of 75 s was selected, which corresponded to the time needed for the loading machine to reach this maximum value of 5 kN.

### 2.2. Simulation the Wear with 1 mm and 2 mm Depth Swept Cut

To simulate the fundamental problem of the wear discussed above and obtain an effective strain profile of the metallic crossing part, a 3D CAD model (150 × 70 × 50 mm) was developed with a swept cut of 40 × 20 mm, and an extruded cut with two depths, 1 mm and 2 mm, successively, as shown in Figure 4a. The dimension of the 3D CAD model corresponded to the actual metallic part used in the experimental tests, resembling the frog of the S&C system. Moreover, the actual metallic part used for the loading test, described at the previous section (Section 2.2), was cut as shown in Figure 4b. The extruded cut dimension corresponded to the minimum cut that the cutter machine used at the university’s workshop. Therefore, Test 0 represents time-dependent loading without the cut, Test 1 represents time-dependent loading with the cut (40 × 20 × 1 mm) and Test 2 represents time-dependent loading with the cut (40 × 20 × 2 mm). The 3D CAD model in Figure 4b with 40 × 20 mm swept cut is highlighting mesh, constraints, and applied load. The extruded cut mesh details are as follow: (i) for 1 mm depth, there are 11,393 nodes, 7315 elements, average element size 5.45 mm, and tetrahedral mesh of high quality; and (ii) for 2 mm depth, there are 11,395 nodes, 7334 elements, average element size 5.45 mm, and high-quality tetrahedral mesh. The part was fixed as indicated by the green arrows in Figure 4a and the load was applied perpendicular to the surface replicating the laboratory environment.

### 2.3. Experimental Linear Loading Tests

The performance of the FBG sensors glued to the fixed frog was evaluated through several tests and strain profiles were generated using Micron Optics Enlight software. As seen in Figure 5, three FBG sensors were placed on the fixed frog to measure the strain produced by the application of the said load. The location on the structure’s surface was carefully selected by looking at the results obtained from the simulation that indicated the areas of interest, according to the Von Mises stress. One sensor (FBG1) was placed very close to the contact with the load where the model showed higher sheer stress. The two other sensors were placed at a distance from the contact to study the strain variation in a relative safer place from the contact with the wheel for easy installation in-the-field. With the relationship between the Elastic Modulus *E*, the stress *σ*, and the strain *ε* being *E* = *σ*/*ε*, and knowing that *E* depends on the structural material and is constant, it can be said that the areas showing higher stresses will also correspond to the areas of higher strains. As seen, FBG1 was placed on the transition zone (clear blue), whereas FBG2 was located on a place where the stress was already settled. As mentioned FBG3 location is just along the part. The total deformation was also checked.

Figure 6 shows the set-up used for the strain measurement using an FBG interrogator and a 30 kN universal strength tensile testing machine controlled by a computer.

As shown in Figure 2b, the frog will be under two forces from the top (wheel) and from the side (the wheel flange). In a real case scenario, the impact will be perpendicular at the top of the frog. To detect the strain generated by this impact, FBG sensors need to be placed at the top of the frog, which is less practical, unless embedded inside a groove to be protected. On the other hand, the side of the frog will be under the applied force from the wheel flange. Therefore, to monitor the wear in this study, controlled linear loads were used with a defined cut (rectangular shape with constant dimensions and with two depths) to simulate the wear. The linear load was applied on the top corner of the metallic part and shown in Figure 7. Two types of tests were carried out to study strain measurement profiles for gradual and sudden changes when wheel–rail rolling contact moves along the track during train motion, and for impulsive change when a train wheel track is standing by the crossing part. The ultimate aim for studying the change of the load dynamic in this research is to imitate, as close as possible, the real behavior of the wheel–rail and frog interaction during the normal operation. In the first case, when the train is moving at a slower speed, so the wheel–rail interaction takes a longer time, and the load will approximately be following the first type of load behavior shown in Figure 7. The second scenario is when the train moves at a higher speed, the duration will be shorter, and the frog will be exposed to a sudden change of load.

## 3. Results and Discussion

### 3.1. Analytical Modeling of the Effect of Cut

In this paper, the time-dependent loading simulation model is developed to examine the geometric effects on the metallic part (simulating the frog part of the S&C) that can change its mechanical properties compared to that for the bulky form. A nonlinear study is appropriate in this case as the model’s boundary conditions vary throughout the study. The part was subject to a nominal force of 5kN that was applied over 75 seconds at room temperature. The solver autostepping option was activated and the default value of 0.01 time increment used. This gives the solver a guideline for how big the time step should be and how to automatically adjust it. As it can be observed the studies have been solved using 6–7 steps. Figure 8a displays the equivalent strain result from the transient structural simulation for the component without the cut. With the equivalent strain, one can describe the state of the strain in the structural material. The equivalent strain, “ESTRN”, is defined by [34], using Equation (1):(1)$$ESTRN=2{\left[\left(\varepsilon_{1}+\varepsilon_{2}\right)/3\right]}^{1/2}$$
where:$$\varepsilon_{1}=0.5\left[{\left(EPSX-\varepsilon^{*}\right)}^{2}+{\left(EPSY-\varepsilon^{*}\right)}^{2}+{\left(EPSZ-\varepsilon^{*}\right)}^{2}\right]$$
$$\varepsilon_{2}=\left[{\left(GMXY\right)}^{2}+{\left(GMXZ\right)}^{2}+{\left(GMYZ\right)}^{2}\right]/4$$
$$\varepsilon^{*}=\left(EPSX+EPSY+EPSZ\right)/3$$
with EPSX being the normal strain in the X-direction of the selected reference geometry, EPSY the normal strain in the Y-direction of the selected reference geometry, and EPSZ the normal strain in the Z-direction of the selected reference geometry. GMXY is the shear strain in the Y direction in the YZ-plane of the selected reference geometry, GMXZ. is the shear strain in the Z direction in the YZ-plane of the selected reference geometry, and GMYZ is the shear strain in the Z direction in the XZ-plane of the selected reference geometry [34]. As seen in Figure 8, Figure 9 and Figure 10, the maximum equivalent strain for the three cases took place at the point of loading application (see purple arrows and element highlighted in red). Plots shown in part (b) of the mentioned Figure 8, Figure 9 and Figure 10 display how the equivalent strain varies over time at the loading application point. It can be observed that the obtained shape corresponds to the one of the load.

Figure 9 and Figure 10 display the equivalent strain results from the transient structural simulations for the components with the 1 and 2 mm cuts, respectively.

The simulation results of the equivalent strain changes over time and show a correlation with the effect of the introduced cut. The equivalent strain seems to increase due the cut effect on the metallic part. The maximum value of the total strain was 650 με without the cut, then increased to 820 με, then 1200 με for the 1 and 2 mm cuts, respectively.

### 3.2. Experimental Strain Measurement

Strain variations from the three sensors FBG1, FBG2, and FBG3 under the two different loadings are illustrated in Figure 11, Figure 12 and Figure 13. Nearly the same profiles of strains were observed for both loading types previously shown in Figure 7. Hence the deviation of strain gauge in reply to sudden change introduced by the wheel-track will not affect the results significantly. For this linear case, with different loading conditions, strain varies smoothly and linearly. 

Sensor FBG1 is the nearest sensor to the cut. As can be seen in Figure 11, the maximum value of strain, which corresponds to the maximum load of 5 kN, changed significantly from 48 με before the cut to less than 8 με when loading the part with the cut. However, the sensor shows quite similar strain measurements for both tests with 1 and 2 mm cuts.

Figure 12 shows the strain measurement from FBG2 sensor. This sensor was placed at a distance of 50 mm from the top of the part to avoid a direct impact with the wheel, and for easy installation in situ. This sensor shows a maximum strain value of 23 με before the cut. This value decreased to 14 με and to 8 με for the tests with cuts of 1 and 2 mm, respectively. It was noticeable that the FBG2 sensor showed some vibration signals, but did not affect the overall strain distribution. From this sensor, the location and the direction of the sensor had a key role in wear detection and its accuracy. Setting the sensor as close as possible to the wear location, similar to FBG1, does not guarantee the results. FBG2 was located at a 40 mm distance from the vertical center of the wear, with an offset of 20 mm compared with FBG1, but the response of the following one was reflective of the overall experiment. The strain level was reduced due to the cut introduced to the metallic part.

Figure 13 shows the strain measurement from the FBG3 sensor. This sensor was placed 50 mm away from the FBG2 sensor, as shown in Figure 5, to study the strain profile along the part. This sensor showed a maximum strain value of 70 με before the cut. This value increased to 88 με and to 85 με for the tests with cuts of 1 and 2 mm, respectively. This sensor was very close to the clamps used in this experimental test to immobilize the part. The cut seems to introduce an increase in strain measurement, around 15 με, with very low effects, between the 1 and 2 mm cuts. Overall, FBG sensors showed correlation with the loading profiles in all experimental tests. However, it seems that the contact point with the loading machine, and the fixing point with the clamps, introduced small vibrations, which helped generate nonlinearities in all sensors.

### 3.3. Comparison between Modeling and Experimental Strain Measurements

Three points were selected from the 3D drawing schematic of the metallic part that roughly corresponded to the same position of the FBG sensors used for the experimental loading tests. Figure 14 illustrates the equivalent strain for the *x*-axis calculated for these points. Figure 14a, shows the strain calculation for the part without the cut. Figure 14a,b show the strain calculation for the 1 and 2 mm cuts, respectively. So, for example, the FBG1 has maximum values of 59.4, 34.7, and 32.6 με compared to the experimental maximum values of 48, 8, and 9 με.

Figure 15 depicts a comparison of maximum strain measurements between simulation models calculated by the software, as shown in Figure 14, and experimental loading tests, shown in Figure 11, Figure 12 and Figure 13. As can be seen in Figure 15, the FEM method provides reasonable estimates of the maximum strain values for FBG1 and FBG2 sensors with a standard deviation of approximately ±20 με. However, the standard deviation seems to be higher for FBG3 for approximately 150 με. Despite the fact that there are some variations between FEM and experimental results, the objective of this research is to study the effect of wear on strain measurements. It is clear from all sensors, and from simulation and experimental tests; there is a correlation between the wear (represented in this paper as a cut) and the strain variations. However, the quantification of the size of the wear needs more lab tests to fully understand this correlation.

## 4. Conclusions

FBG sensors have been measuring strains for decades. However, the novelty of this paper was to study the potential of these sensors to monitor the wear on railway S&C. The wear effect on railway S&C systems is extremely difficult to monitor and to quantify. Therefore, FBG sensors can offer an innovative solution by providing information about the behaviors (e.g., strain measurement) of the S&C systems under different loading conditions.

The strain profiles from three FBG sensors, placed on metallic parts similar to railway crossings, were measured to evaluate sensor responses to predefined cuts, and their accuracies for detecting discontinuity of railway paths. The sensors were examined under periodic and controlled loadings to determine if wear-and-tears, simulated as cuts, affected the strain measurement. The study also verified that precise locations of the initial strain, as well as strain build-up, can be identified in conjunction with early failure warnings. A numerical model of a 3D prototype was developed through the use of a finite element method piece of software, in order to define loading testing conditions and sensor locations, as well as for comparison with experimental tests.

Data retrieved from the simulation studies showed that the strain changed when modifying the geometry of the 3D model (a cut was introduced to simulate the wear effect). However, the simulation results show higher strain values compared to the experimental results, especially for the FBG3 sensor, which was far from the loading point and near to the clamps used to fix the part. The higher values in the simulation, which was based on a numerical modeling process, could be affected by the hypotheses in the model, as well as by further software resolution errors. It was observed that the location of the FBG2 represents a safer location to install such sensors without jeopardizing the monitoring process. However, the location of the sensor near to the top edge of the frog (e.g., the location of FBG1 sensor) can show a better understanding of the variation of the strain profile due to the wear.

From the FEM study, and experimental laboratory tests results, the FBG sensor is a potential candidate that can be used for health monitoring of switches and crossings. To the best of our knowledge, no previous research that estimates the wear on the S&C using FBG sensors has been conducted. Based on simulation results within this pilot study, we have the basic theory on how to implement FBG sensors as a condition monitoring system for the crossing part of the S&C. This study has provided preliminary results, using only simulation and laboratory experimental tests, to show that strain measurements have changed with the introduced cut on the metallic piece (representing the wear effect on the frog of the S&C) under very low loading conditions (5 kN). This article is opening new grounds, in in-field solutions, to monitor railway S&C. We are hopeful that this article will create further research studies to help establish more tests with heavier applied loads and field tests.

## Figures and Tables

**Figure 1 sensors-21-03639-f001:**
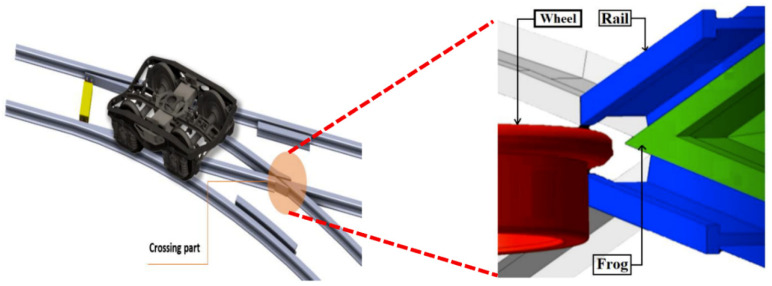
Railway switches and crossings.

**Figure 2 sensors-21-03639-f002:**
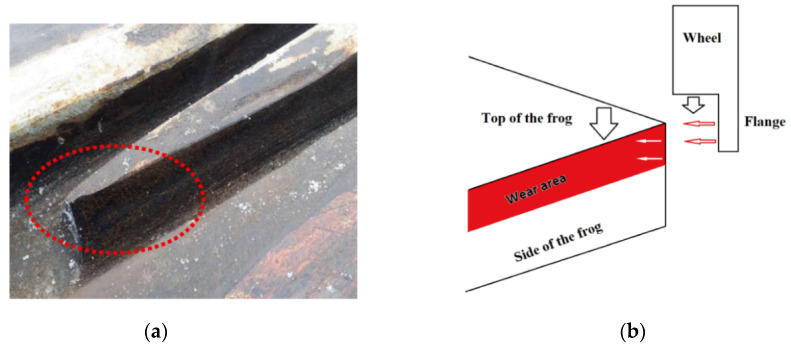
(**a**) The wear effects on the side of the frog in railway switches and crossings; (**b**) a schematic figure to describe the applied loading forces from the wheel and the wheel flange on the frog.

**Figure 3 sensors-21-03639-f003:**
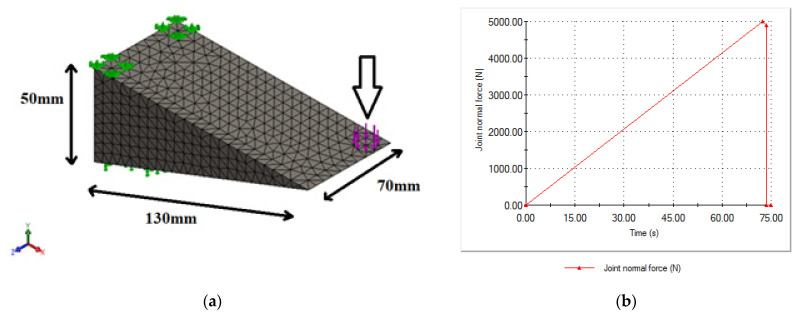
(**a**) Three-dimensional (3D) meshed model and (**b**) applied force used in the simulation model.

**Figure 4 sensors-21-03639-f004:**
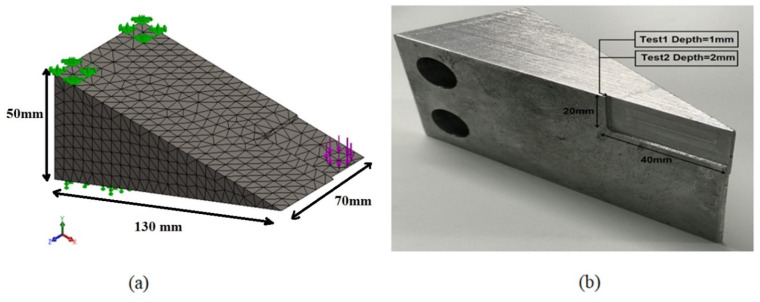
(**a**) The 3D drawing schematic used for the simulation study and (**b**) the actual test set-up representing the cut for simulating the wear and used for the experimental strain measurement tests. The two holes are needed to fix the metallic part; all parts are bolted to a large steel bed secured to heavy steel support.

**Figure 5 sensors-21-03639-f005:**
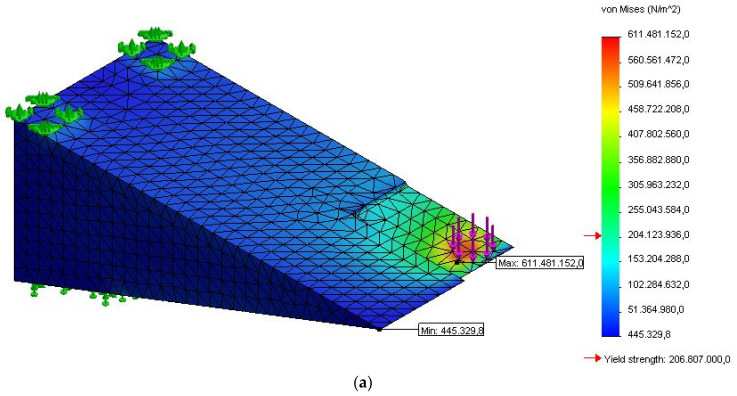
(**a**) Three-dimensional (3D) CAD model highlighting Von Mises stress for identification of sensing elements location; (**b**) FBG sensors location; (**c**) experimental set-up used for loading test.

**Figure 6 sensors-21-03639-f006:**
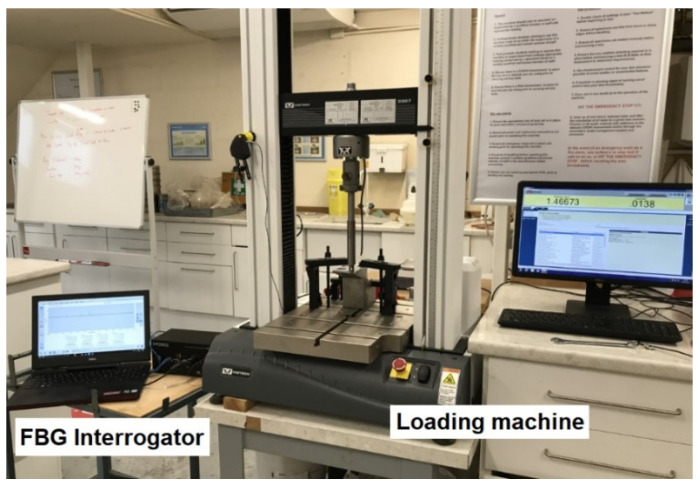
The set-up used for the experimental strain measurement tests.

**Figure 7 sensors-21-03639-f007:**
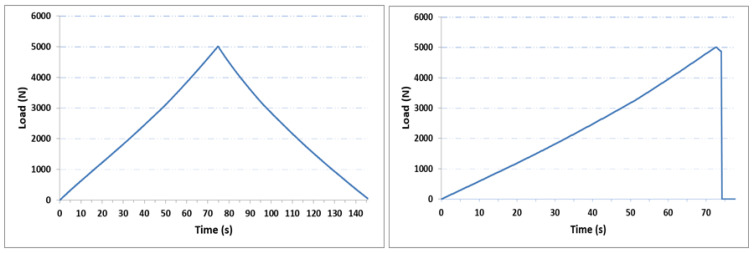
Load profiles generated by the Instron 3367 30 kN Universal Strength Tensile Testing. **Left**: gradual changes; **Right**: sudden changes.

**Figure 8 sensors-21-03639-f008:**
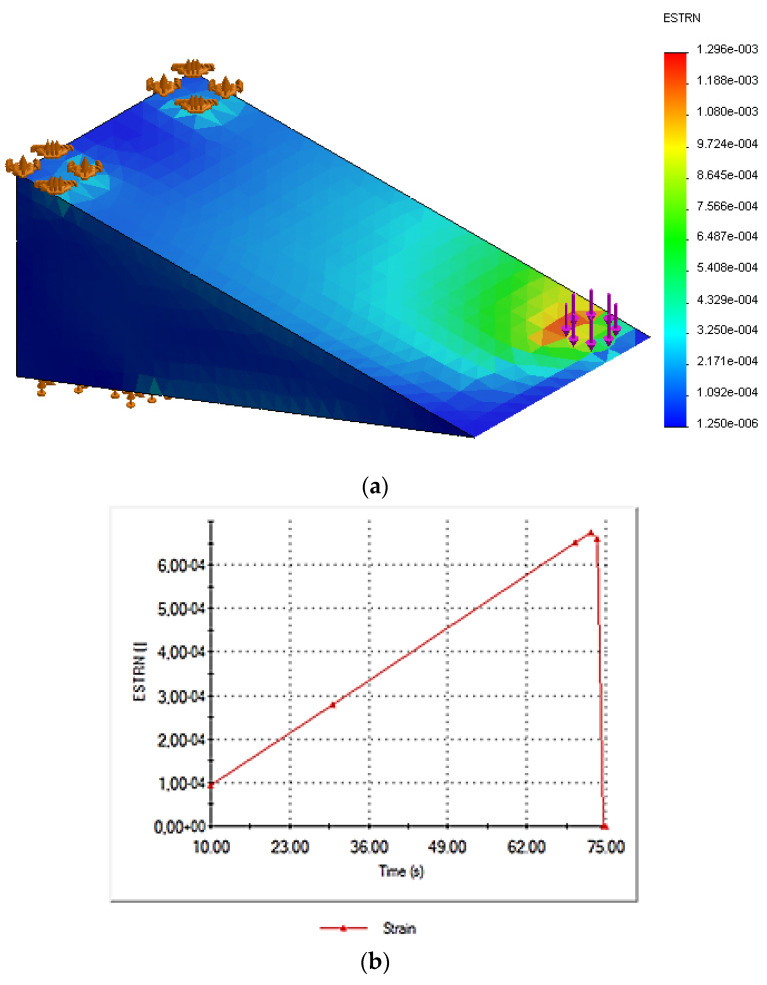
(**a**) Three-dimensional (3D) model highlighting equivalent strain; (**b**) equivalent strain vs. time for the model without the cut.

**Figure 9 sensors-21-03639-f009:**
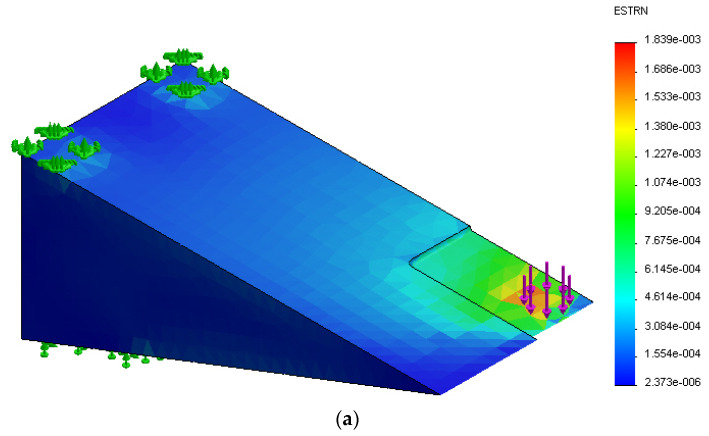
(**a**) Three-dimensional (3D) model highlighting equivalent strain; (**b**) equivalent strain vs. time for the model with 1 mm cut.

**Figure 10 sensors-21-03639-f010:**
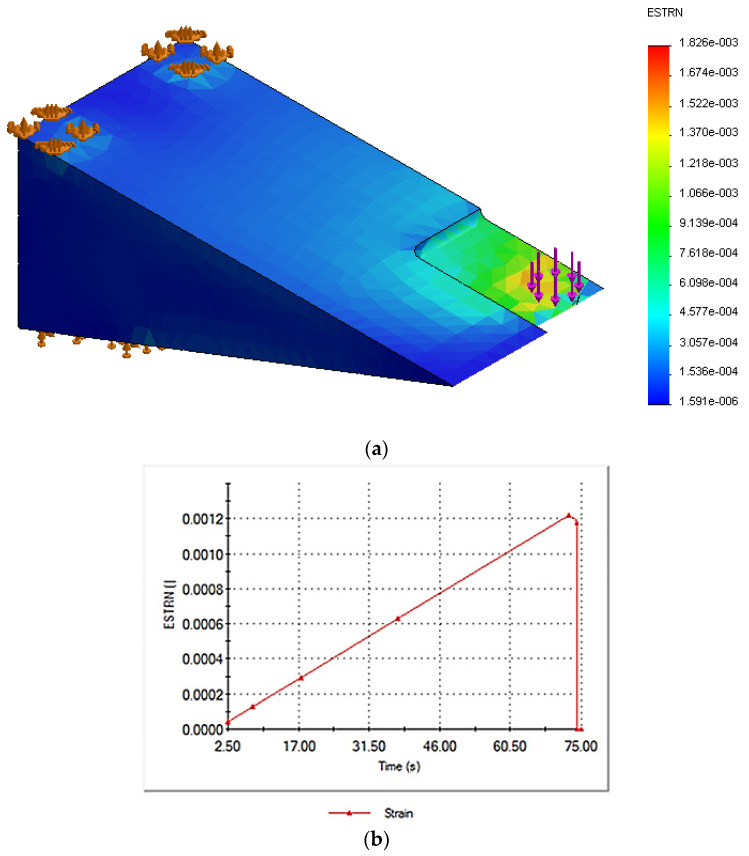
(**a**) Three-dimensional (3D) model highlighting equivalent strain; (**b**) equivalent strain vs. time for the model with 2 mm cut.

**Figure 11 sensors-21-03639-f011:**
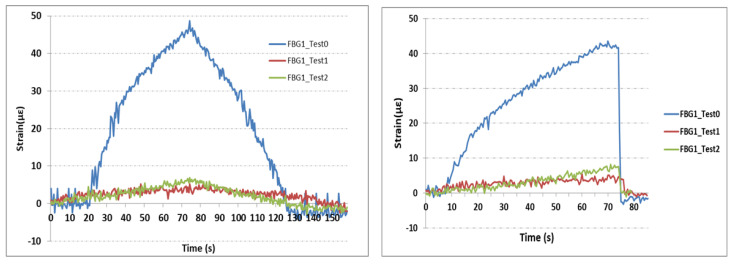
Strain measurements from FBG1 sensor. **Left**: Strain measurements from the test profile of gradual changes; **Right**: Strain measurements from the test profile of sudden changes.

**Figure 12 sensors-21-03639-f012:**
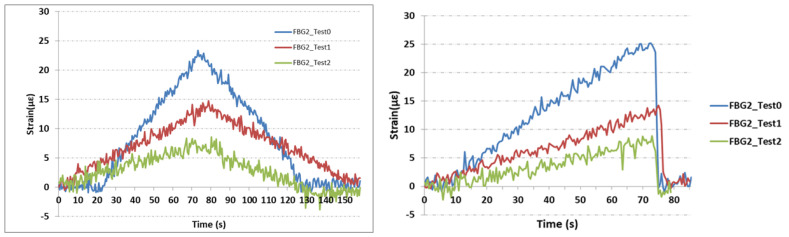
Strain measurements from the FBG2 sensor. **Left**: Strain measurements from the test profile of gradual changes; **Right**: Strain measurements from the test profile of sudden changes.

**Figure 13 sensors-21-03639-f013:**
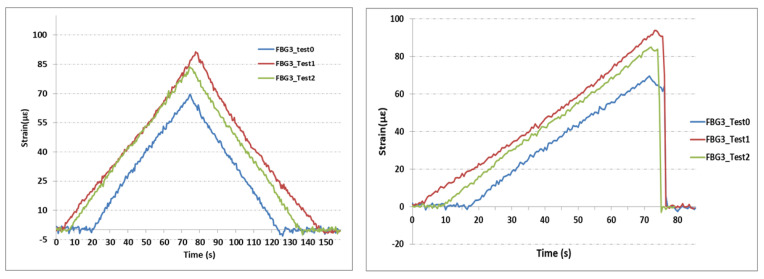
Strain measurements from the FBG3 sensor. **Left**: Strain measurements from the test profile of gradual changes; **Right**: Strain measurements from the test profile of sudden changes.

**Figure 14 sensors-21-03639-f014:**
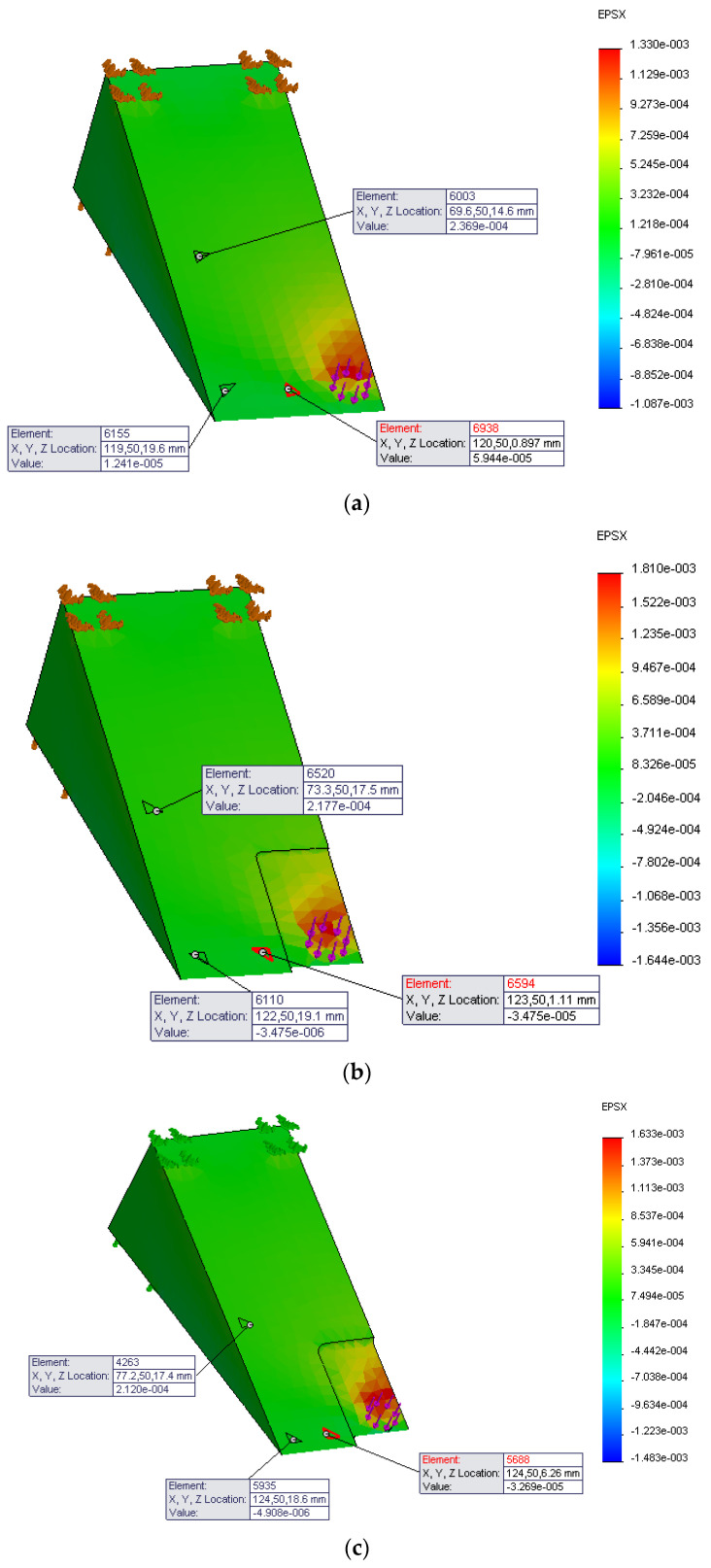
The equivalent strain calculated at three points from tests (**a**) without cut, (**b**) with 1 mm cut, and (**c**) with 2 mm cut.

**Figure 15 sensors-21-03639-f015:**
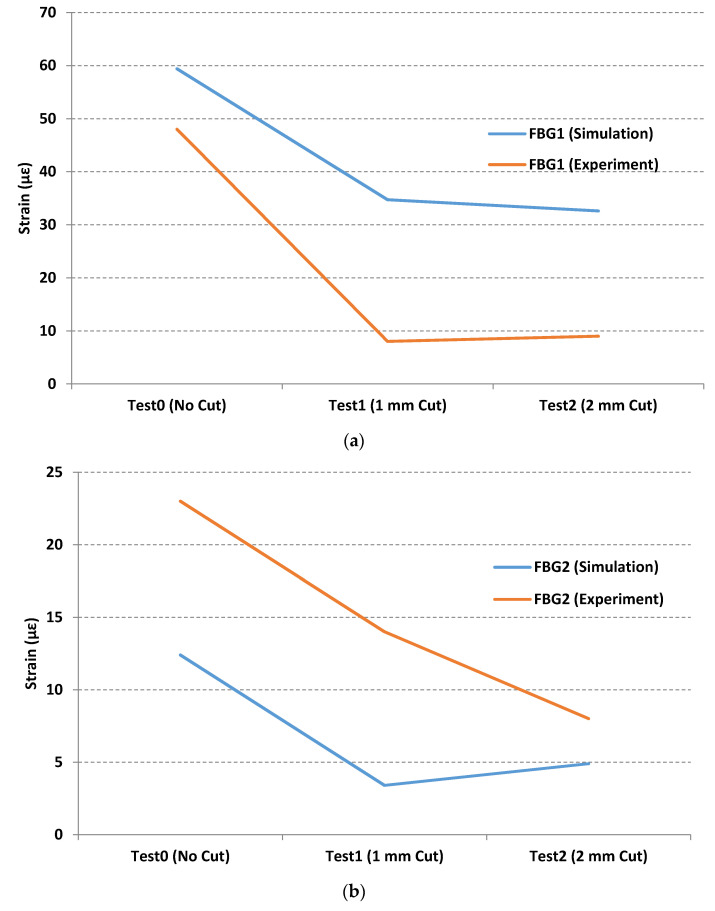
Comparison of maximum strain measurements between simulation and experimental tests for (**a**) Sensor FBG1; (**b**) Sensor FBG2 and (**c**) Sensor FBG3.

## Data Availability

Not applicable.

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
