# Peer review of "Experimental Strain Measurement Approach Using Fiber Bragg Grating Sensors for Monitoring of Railway Switches and Crossings"

_sensors, 2021, doi:10.3390/s21113639_

Round 1

Reviewer 1 Report

The proposed manuscript is about monitoring railway wear state using fiber Bragg gratings focusing on switch and crossing. Generally, the main problem to the reviewer is the lacks of the creativity. Fiber Bragg gratings has been shown to be able to detect static strain by several researchers. The reviewer points out some concerns as follows, not in a logical order.

  • First, there are several redundant statements in the introduction. For example, in the last paragraph, the authors mentioned optical fiber sensors, including FBGs, then in the first paragraph, the authors described FBGs again with a lot of similar and looks like redundant statements. In fact, even when it comes to section 2, methods and measurement, the authors described FEMs in the approach of writing introductions. Generally, the whole manuscript exists a lot of redundant or seemly redundant statements.
  • What is 2 steps applying loadings over 75 seconds. How and why 2 steps?
  • Why choosing linear loading? In real scenario, wouldn’t it be impact loading or moving loadings?
  • The dimensions of the simulated structure are not provided in sec 2.1.
  • The loading and boundary conditions are not clearly stated in Fig. 2(a) and Fig. 3(a). Also, the loading vectors are not clearly shown in the figures, where they should be perpendicular to the surface?
  • What are the two holes in Fig. 3(b).
  • How did the authors choose the location of the FBGs. In real scenario, the locations of the wears are not known beforehand.
  • Why are the two loadings different in sec 2.1 and sec 2.3. Should the authors consider alternative loadings?
  • Why, in the 2mm cut case, don’t the strain fall to zero?
  • Why negative strain shown in Fig. 10? Why is the loading different in Fig. 9?
  • Why are there shift of maximum strain?
  • The main problem is that/might be that the relation of the wears and the strain variations is not clearly addressed, which makes the simulation and experiments less useful.
  • The quantity of Young’s modulus is not correct.
  • About Fig. 6, the loading curve does not agree to the real situation, where the loading from train will keep a near constant value before dropping off?
  • In section 3.2, the results obtained from the FBG sensors are not clearly explained.
  • A lot of typos or similar problems, for example, Fiber “Brag” Grating… “war” size…
  • What is “where-and-tears”?

Reviewer 2 Report

In the manuscript entitled "Experimental strain measurement approach using Fiber Bragg Grating sensors for railway switch & crossing monitoring " [Manuscript ID sensors-1187423] submitted to the MDPI Sensors journal, the authors described their proposed method for monitoring excessive wear on the frog of a railway switch and crossing (S&C) using fiber Bragg grating (FBG) sensors. The FBGs are utilized to monitor the strain profile, based on which the frog wear size is assessed. The proposed solution is evaluated theoretically using a finite element analysis and experimentally under periodic and controlled loading tests. The experimental and numerical results are considered to be in good agreement.

The authors are encouraged to address the following concerns and suggestions to improve the quality of this contribution:

  1. In the title, line 3, the ‘&’ should be replaced with the ‘and’ word.
  2. Line 34, references should be listed according to the journal requirements.
  3. Line 56, “… the war size”. Should it be “the wear size”?
  4. Figure 1, what is the origin of the presented graphics? Please provide the relevant reference, if appropriate. The quality of the figure should also be improved.
  5. Line 74, the statement “Failure of the frog point as shown in Fig1,” should be amended as it might be misunderstood by the reader. The failure is not clearly seen in the figure.
  6. Line 105, the sentence “… optical fiber sensors (such as Mach-Zehnder, Michelson…” should be clarified to the readers. What are the Mach-Zehnder and Michelson sensors? Please provide the relevant references.
  7. Lines 107-108, “These sensors are well known…[17and 18].” The statement is inaccurate and needs to be reworded especially when the remarks from point 6 are considered. Please provide more references supporting this statement.
  8. Lines 112-114, please provide references to support the sentence “many technologies for monitoring railway … or image processing.”
  9. Lines 125-127, “The objective… to explore the viability of FBG based sensing mechanisms to monitor railway systems to act as an early failure warning device.” The FBG sensing mechanisms are well known and are not related to railway systems. This sentence should be reworded.
  10. Line 147, “lab” should be replaced with “laboratory”.
  11. Line 148, “at room temperature”, please specify what was the room temperature.
  12. Lines 180-181, 242, please explain why such conditions were chosen? Why 5 kN was applied over 75 seconds? Please justify the used force and time. Please specify what room temperature was.
  13. Lines 188-190, please justify what was the reason for choosing the specified dimensions for the sample size and the cut depths.
  14. Line 247, please explain what “the equivalent strain” is and how each plot shown in (b) is related to the sample geometry (a) presented in Figures 7-9? Please mark the relevant point on the geometry.
  15. Figure 10-12, please explain the reasons for the FBG sensor response nonlinearities.
  16. Line 324, please explain the differences between “standard variation” and “standard deviation”. How is the standard variation defined?
  17. Line 336, “A new measuring method of where-and-tears…” Please clarify what the new measuring method presented in the paper is. Strain measurement using an FBG is not new. If the application is new, please clearly state it. Also, “where-and-tears” should be “wear and tear”, I believe? Please correct as required.
  18. Lines 346-349, “Data retrieved from the simulation studies showed a good match with the values obtained from the experiments. It was observed…” The differences between the simulations and the measurements, shown in Figure 14, are over 100% in some cases which is significant. This statement needs to be amended accordingly as it is in contradiction to the presented results.
  19. Lines 358-359, “The most significant finding is that it is possible, in a real scale test rig to indirectly and accurately measure the wear size.” This has not been shown or proven. The accuracy of the proposed measurement method was not evaluated appropriately, and the wear size was not estimated, especially that in lines 329-330, the authors state: “However, the quantification of the size of the wear needs more lab tests to fully understand this correlation.” Please amend this statement accordingly or delete the sentence.
  20. Please correct the paper formatting according to the journal template.
  21. Please check the paper against typo errors and make the relevant corrections.

Reviewer 3 Report

This manuscript presented strain measurement approach using Fiber Bragg Grating sensors for railway switch & crossing monitoring. The results are interesting. 1) What kinds of FBG were used for the experments, such as central wavelength and reflectivity. 2) How was it decided to use three FBGs? 3) In comparison of maximum strain measurements between simulation and experimental tests (Figures 14(b and c), there are obvious rend differences. What are the reasons?

Round 2

Reviewer 1 Report

The authors have improved the manuscript.

Reviewer 2 Report

My comments and suggestions have been satisfactorily addressed by the authors.